# Synchrony-Division Neural Multiplexing: An Encoding Model

**DOI:** 10.3390/e25040589

**Published:** 2023-03-30

**Authors:** Mohammad R. Rezaei, Reza Saadati Fard, Milos R. Popovic, Steven A. Prescott, Milad Lankarany

**Affiliations:** 1Krembil Research Institute, University Health Network (UHN), Toronto, ON M5T 0S8, Canada; 2Institute of Biomedical Engineering, University of Toronto, Toronto, ON M5S 3G9, Canada; 3KITE Research Institute, Toronto Rehabilitation Institute, University Health Network (UHN), Toronto, ON M5G 2A2, Canada; 4Department of Computer Science, Worcester Polytechnic Institute, Worcester, MA 01609, USA; 5Department of Physiology, University of Toronto, Toronto, ON M5S 1A8, Canada; 6Neurosciences and Mental Health, The Hospital for Sick Children, Toronto, ON M5G 1X8, Canada

**Keywords:** neural coding, information representation, multiplexed coding, synchronous and asynchronous spikes, general linear model

## Abstract

Cortical neurons receive mixed information from the collective spiking activities of primary sensory neurons in response to a sensory stimulus. A recent study demonstrated an abrupt increase or decrease in stimulus intensity and the stimulus intensity itself can be respectively represented by the synchronous and asynchronous spikes of S1 neurons in rats. This evidence capitalized on the ability of an ensemble of homogeneous neurons to multiplex, a coding strategy that was referred to as synchrony-division multiplexing (SDM). Although neural multiplexing can be conceived by distinct functions of individual neurons in a heterogeneous neural ensemble, the extent to which nearly identical neurons in a homogeneous neural ensemble encode multiple features of a mixed stimulus remains unknown. Here, we present a computational framework to provide a system-level understanding on how an ensemble of homogeneous neurons enable SDM. First, we simulate SDM with an ensemble of homogeneous conductance-based model neurons receiving a mixed stimulus comprising slow and fast features. Using feature-estimation techniques, we show that both features of the stimulus can be inferred from the generated spikes. Second, we utilize linear nonlinear (LNL) cascade models and calculate temporal filters and static nonlinearities of differentially synchronized spikes. We demonstrate that these filters and nonlinearities are distinct for synchronous and asynchronous spikes. Finally, we develop an augmented LNL cascade model as an encoding model for the SDM by combining individual LNLs calculated for each type of spike. The augmented LNL model reveals that a homogeneous neural ensemble model can perform two different functions, namely, temporal- and rate-coding, simultaneously.

## 1. Introduction

Transmitting multiple signals over a single communication channel increases the channel bandwidth and enhances the coding efficiency [1,2]. Similar to digital communication systems, the brain utilizes different forms of multiplexing—in different brain regions and in regard to different stimuli—to represent multiple features of a stimulus with a neural code [2]. For example, in the auditory sensory system, the frequency and intensity of a periodic stimulus are encoded by the phase-locked spikes and the probability of spiking per stimulus cycle, respectively [3]. Similarly, the frequency and intensity of vibrotactile stimuli are represented by the timing and rate of spikes in the somatosensory cortex [4]. Recently, differentially synchronized spiking in neurons of the primary somatosensory cortex was shown to enable multiplexed coding of low- and high-contrast features of tactile stimuli [5].

Despite various forms of neural multiplexing, a thorough understanding of how the brain achieves multiplexing remains undiscovered. Specifically, the functional characteristics—in the sense of linear or nonlinear filtering properties—of a neural ensemble that multiplexes different features of a stimulus are yet to be uncovered.

Different features of stimuli, such as the intensity, frequency, onset and offset, etc., dictate which multiplexing strategies are the most appropriate [4]. In addition to the properties of stimuli, heterogeneity of the neurons in a population code enables different neurons to encode different stimulus features. The functional properties of a heterogeneous neural ensemble, which includes neurons with different functions, e.g., integrators vs. coincidence detectors, might be fully described by the dynamics of individual neurons. For example, an ensemble of heterogeneous cochlear nuclei in the auditory cortex is composed of two anatomically distinct sub-nuclei, namely, the magnocellular and the angular nucleus, each of which selectively encodes a specific feature of the stimulus. The magnocellular nucleus selectively encodes the stimulus frequency with a temporal code by implementing a high-pass filter, whereas the angular nucleus selectively encodes the stimulus intensity with a rate code by implementing a low-pass filter [6]. In contrast to heterogeneous neural ensembles, the functional characteristics of an ensemble of homogenous neurons, which includes neurons with nearly identical functions, cannot be identified based on the properties of individual neurons solely [4,5]. For example, in synchrony-division multiplexing (SDM) [5], information about slow and fast stimulus features were respectively represented by asynchronous and synchronous spikes of the same neurons. Thus, this form of multiplexing suggests that both slow and fast features of the stimulus can be encoded by homogeneous (identical) neurons that operate in a hybrid mode [5], i.e., neither low-pass nor high-pass filtering of the stimulus [7,8]. Thus, a challenging question is whether multiplexing (such as SDM) in a homogeneous neural ensemble reveals system-level functions beyond those performed by individual neurons [5].

In this paper, we utilize conductance-based and linear nonlinear (LNL) cascade models to establish a theoretical framework to address this question [9,10,11,12]. First, we use conductance-based models and construct a homogeneous neural ensemble that multiplexes the slow and fast features of a common stimulus using asynchronous and synchronous spikes, respectively. Using LNL cascade models, we explore whether different linear filters and static nonlinearities are associated with different types of spikes. We show that a low-pass filter followed by a nonlinearity with a mild slope generates asynchronous spikes whereas a high-pass filter followed by a nonlinearity with a steep slope detects fast features of the stimulus by generating synchronous spikes. Then, we develop an augmented LNL model for SDM by integrating the LNL models underlying each type of spike.

## 2. Results

In the present paper, we developed an augmented LNL cascade model as an encoding model for the SDM [5]. Conductance-based neuron models were used to create an ensemble of homogeneous neurons whose input (mixed stimulus)–output (spikes) relationship was estimated with the augmented LNL model.

As shown in Figure 1A, we construct an ensemble of homogenous neurons with 30 Morris–Lecar (ML) neuron models (see Methods), all of which receive a common mixed signal comprising slow and fast features as well as independent, physiologically realistic conductance noise [5,13]. We divided the simulated data (20 s of data) into two training (the first 50% of data) and test (the remaining 50%) sets. The parameters of the ML model were selected in a way that all neurons operate in a hybrid mode [14]. Spikes generated by an ensemble of ML neurons were used to fit the augmented LNL model in which two separate LNL models were combined to represent the rate and temporal codes simultaneously. This study shows that an ensemble of homogeneous neurons utilizes different strategies to generate synchronous and asynchronous spikes, which enable the simultaneous coding of fast and slow features of a mixed stimulus, respectively. Although the biophysical mechanisms underlying implementation of SDM with an ensemble of homogeneous neurons is still unknown, the two-stream augmented LNL model provides a system-level understanding of SDM function.

### 2.1. Different Temporal Filters Map Distinct Features of a Mixed Stimulus

To explore how the slow and fast features of the stimulus are encoded by spikes of an ensemble of neurons, we used well-known feature space estimators such as the spike-triggered average (STA) [15,16] or information-theoretic spike-triggered average and covariance (iSTAC) to reveal the temporal characteristics of neurons in response to a stimulus [17].

The STA filter is a precise and unbiased predictor for a neural population given a stationary and single-dimension stimulus [16]. However, it fails to provide precise predictions when the dimensionality of the stimulus is larger than one. For example, in retinal ganglion cells the STA cannot predict the neural response of both ON and OFF cells given a mixed input comprising more than one feature. To explore other possible subspace features of the neural response, we used the iSTAC method and calculated the optimal subspace features. The iSTAC quantifies the significance of subspaces based on the mutual information between the stimulus and neural response [17]. In this method, we choose the eigenvectors of the spike-triggered stimulus ensemble matrix more precisely by minimizing the Kullback–Leibler (KL) divergence between the eigenvectors of the ensemble matrix and the raw stimulus distributions (see Methods for more details). In fact, the iSTAC maximizes information based on the first two moments of the spike-triggered stimulus ensemble and provides a unifying information-theoretic framework that captures the ensemble neuron activity in different subspaces. This provides an implicit model of the contribution of the nonlinear function mapping the feature space to the neural response. As shown in Figure 1B (left), the iSTAC matrix calculated for the mixed stimulus has two significant eigenvalues whose underlying eigenvectors reveal two distinct temporal filters, namely, $\nu_{1}$ and $\nu_{2}$. The projection of the spike-triggered stimulus ensemble on $\nu_{1}$ and $\nu_{2}$, shown in Figure 1B (right), reveals two distinct clusters.

In synchrony-division multiplexing, a mixed-input signal containing slow and fast features drives an ensemble of neurons. The fast component of the stimulus whose neural representation is synchronous and sparse does not appear in the STA, as STA averages out the sparsely-occurring fast features of the stimulus [5]. However, unlike the fast signal, the neural representation of the slow signal is asynchronous and dense; thus, the STA filter mainly contains information of the slow components of the mixed signal [5]. Unlike the STA filter, the most informative subspaces selected by the iSTAC method behave as multi-space feature estimators and illustrate the slow and fast features of the mixed stimulus.

Figure 1C shows that the STA filter calculated for the mixed stimulus mainly captures the slow feature of the signal but cannot truly capture the dynamics of synchronous spikes. Unlike the STA filter, $\nu_{1}$ and $\nu_{2}$ of the iSTAC method illustrate the slow and fast features of the mixed stimulus, respectively. As can be observed in Figure 1C, $\nu_{1}$ is similar to the STA filter and represents the slow component of the stimulus, and $\nu_{2}$ describes the fast features of the stimulus (note that the STA filter was duplicated in Figure 1C (left and right) and compared with both $\nu_{1}$ and $\nu_{2}$).

### 2.2. Low-Dimensional Feature Space of the Neural Response Can Be Characterized by the STAs of Synchronous and Asynchronous Spikes

Recently, it has been shown that synchronous and asynchronous spikes encode information about the fast and slow features of a mixed stimulus (equivalent to that used in the present study), respectively [13,18,19]. Using an information-theoretic approach, it was shown that synchronous and asynchronous spikes carry information in different time scales. By classifying the spikes of a population of neurons into synchronous and asynchronous spikes, it was demonstrated that the STA filters underlying these spikes, namely, $\mu_{Async}$ and $\mu_{Sync}$, reflect the fast and slow features of the stimulus, respectively. Figure 2A shows the classified synchronous (red) and asynchronous (blue) spikes in the raster plot.

Here, we compared these filters with those obtained using the iSTAC method. First, we tested if the projection of the spike-triggered stimulus ensemble on $\mu_{Async}$ and $\mu_{Sync}$ creates two distinct clusters similar to that projected on $\nu_{1}$ and $\nu_{2}$. As shown in Figure 2B (left), two distinct and separable clusters were generated by $\mu_{Async}$ and $\mu_{Sync}$. More importantly, one can distinguish between these clusters by projecting the synchronous- and asynchronous-spike-triggered stimulus ensemble on $\mu_{Async}$ and $\mu_{Sync}$. Figure 2B (right) reveals that these stimulus ensembles are separable and mutually exclusive. Figure 2C shows the temporal patterns of $\mu_{Async}$ and $\mu_{Sync}$ versus the STA filter. As expected, $\mu_{Async}$ resembles the STA filter, indicating the slow features of the stimulus, and $\mu_{Sync}$ (similar to $\nu_{2}$) describes abrupt changes in the stimulus.

Furthermore, to investigate the functional roles of the above filters, we tested how they contribute to signal reconstruction. The reconstructed signal was obtained by the convolution of spikes—either all spikes for STA (Figure 3A) or $\nu_{1}$ and $\nu_{2}$ (Figure 3B) or asynchronous and synchronous spikes for $\mu_{Async}$ and $\mu_{Sync}$, respectively (Figure 3C). Figure 3 illustrates a 10 s sample of the reconstructed signal using these methods. As is clear in Figure 3B, the signal reconstructed with $\nu_{1}$ and $\nu_{2}$ (iSTAC method) resembles that generated with $\mu_{Async}$ and $\mu_{Sync}$, and both of these signals better capture the fast features than that obtained by the STA filter, indicating the functional relevance between these filters.

### 2.3. Different Nonlinear Functions Are Associated with Synchronous and Asynchronous Spikes

Given different temporal filters underlying synchronous and asynchronous spikes, we sought how these filters map the fast and slow features of the mixed stimulus to the firing rate of an ensemble of conductance-based model neurons. Moreover, since the dynamics of a neural ensemble is not fully linear, these linear filters are not sufficient to project the stimulus to spikes. We utilized a well-known phenomenological model, namely, the LNL cascade model, which uses a linear stimulus filter followed by a static nonlinear transformation, to estimate the firing rate of an ensemble of neurons. Figure 4 and Figure 5 (panel A for both figures) show the LNL diagram for asynchronous and synchronous spikes, respectively. We tested if the linear filter and static nonlinearity are different for synchronous and asynchronous spikes given a common mixed signal. We obtained static nonlinearity functions for synchronous and asynchronous spikes by applying $\mu_{Sync}$ and $\mu_{Async}$ filters to the mixed stimulus (s) and mapping their outputs (through the nonlinearity) to the peri-stimulus time histogram (PSTHs) of the synchronous and asynchronous spikes, respectively:(1a)$$PSTH_{Async}=f_{Async}\;\left(\mu_{Async}*s\right)$$
(1b)$$PSTH_{Sync}=f_{Sync}\left(\mu_{Sync}*s\right)$$
where $f_{Async}\left(x\right)$ and $f_{Sync}\left(x\right)$ are the nonlinearities associated with the asynchronous and synchronous spikes, respectively.

Figure 4B and Figure 5B show, respectively, the raw nonlinearities for asynchronous and synchronous spikes that correspond to the mapping of every single point of the output of the linear filters (*x*-axis) to the values of the PSTHs (*y*-axis). For the nonlinearities underlying the asynchronous and synchronous spikes, we fitted ReLU nonlinearity and sigmoid functions, respectively [20]. The nonlinearity associated with the asynchronous spikes, $f_{Async}\left(x\right)$, has a shallow slope and broad dynamic range, enabling rate-modulated coding. In contrast, the nonlinearity underlying the synchronous spikes, $f_{Sync}\left(x\right)$), has a steep slope and narrow dynamic range, enabling detection of events (i.e., abrupt changes). Although more sophisticated nonlinear functions could provide better fits, we chose simple and well-established nonlinear functions to highlight the difference in the shapes of the nonlinearities underlying rate versus temporal codes in the context of SDM. The instantaneous firing rates of each type of spike can be constructed by passing the output of the temporal filter through the fitted nonlinearities. These firing rates were estimated and drawn against the PSTHs of the asynchronous and synchronous spikes for the test data in Figure 4C,D and Figure 5C,D, respectively. As shown in these figures, the nonlinear functions and estimated PSTHs underlying the temporal filters obtained using the iSTAC (*V1* and *V2*) and classified spikes ($\mu_{Async}$ and $\mu_{Sync}$) are nearly identical.

### 2.4. An Augmented LNL Cascade Model for Synchrony-Division Multiplexing

The LNL cascade models were utilized to encode specific features of a mixed stimulus with synchronous or asynchronous spikes. As shown in the previous sections, temporal filters and nonlinear transformations of either type of spike was distinct and estimated using separate LNL cascade models. Here, we sought whether a combination of these cascade models, i.e., an augmented LNL model, could accurately encode different features of a mixed stimulus through different types of spikes. We developed a two-stream LNL cascade model that combines the PSTHs of the synchronous and asynchronous spikes to reconstruct the mixed PSTH of both types of spikes, as:(2)$$PSTH_{total}=\sum_{i\in \left\{Sync,\;Async\right\}}^{}\;\;\omega_{i}\times G_{i}*f_{i}\left(\mu_{i}*s\right)$$
where $\omega_{i}s$ are the combination weights for each stream of reconstructed PSTHs. To reduce the model complexity and promote smoothness in the output, we applied parameterized Gaussian kernels, $G_{Async}=Gaussian\;\left(0,\sigma_{Async}\right)$ and $G_{Sync}=Gaussian\;\left(0,\sigma_{Sync}\right)$, to the reconstructed PSTHs in each stream [21,22]. Figure 4B and Figure 5B show, respectively, the raw nonlinearities for the asynchronous (f_Async) and synchronous (f_Sync) spikes that correspond to the mapping. The augmented LNL model simultaneously encodes the slow and fast features of the stimulus using asynchronous and synchronous spikes, respectively. Figure 6A shows the block diagram of the augmented LNL model. This model implies that the low-pass filter and shallow non-linearity underlying the asynchronous spikes are required to produce the rate code. In contrast, the high-pass filter and sigmoid nonlinearity for synchronous spikes are necessary to preserve the reliable spike times underlying fast features of the stimulus. Taken together, the augmented LNL model allows rate and temporal to coexist and represent distinct features of the mixed signal. the coexistence of the rate and temporal codes happen to encode multiple features of a mixed stimulus. To capitalize on the significance of temporal filters and nonlinearity transformations of each type of spike in estimating the total firing rate of a neural ensemble, we compared the performance of the augmented LNL model with that of a conventional one-stream LNL. Figure 6B–D shows the firing rate estimated by the three methods, namely, Poisson GLM and augmented LNL models (Figure 6C,D) (see Section 2.3), against the PSTH of ensemble of neurons for the test data. We also quantitatively measured the performance of the augmented LNL and Poisson models in Table 1. As can be observed, the firing rate estimated using the augmented LNL model can better capture both the rate of asynchronous spikes and the timing of synchronous events compared to those estimated using the one-stream Poisson GLM.

## 3. Discussion

The ability of an ensemble of homogeneous cortical neurons to multiplex multiple features of a mixed stimulus was studied in [5]. The specific mechanism by which these neurons encode different features remains to be determined. In this paper, we presented a computational framework to provide a system-level understanding of the encoding mechanism underlying SDM. We used conductance-based neuron models to construct a homogenous neural ensemble that encodes the slow and fast features of a mixed stimulus through asynchronous and synchronous spikes, respectively. To elucidate the contribution of slow and fast features of the mixed stimulus to the spikes generated by the model neurons, we calculated the most significant subspaces (eigenvectors) of the spike-triggered stimulus matrix using the iSTAC method. We demonstrated that the calculated first and second eigenvectors resemble the slow and fast features of the stimulus, respectively. Furthermore, the projection of the spike-triggered stimulus matrix on these eigenvectors created two distinct clusters. We tested whether these clusters can be characterized by synchronous and asynchronous spikes. By computing the spike-triggered average (STA) filters of the synchronous and asynchronous spikes and projecting this matrix on these filters, we clearly separated those clusters. Furthermore, we fitted an LNL model to each type of spike. Similar to distinct temporal filters for synchronous and asynchronous spikes, their static nonlinearities are different. We found that the nonlinearity associated with the asynchronous spikes is very shallow and can be approximated with a linear function. On the other hand, the nonlinearity associated with synchronous spikes has a very large slope and can be approximated using highly nonlinear functions such as the sigmoid function. Finally, we developed an augmented LNL model both to capture the dynamical characteristics of the synchronous and asynchronous spikes and to reconstruct the PSTH of all spikes.

### 3.1. Subspace Feature Extractors: iSTAC vs. STC

To explore more than one subspace feature for stimulation-evoked neural responses, we compared the performance of the STC and iSTAC methods. One can find the most informative subspaces that maximize the mutual information between stimulus and response [23,24]. Nevertheless, an accurate estimation of mutual information requires a large amount of data, although no guarantee for optimal estimation can be necessarily expected [24]. A conventional way to find these subspaces is to calculate those related to the most significant eigenvectors of the spike-triggered covariance (STC) matrix [16]. The eigenvectors of the STC matrix provide analytic expressions for filter estimation using the moments of the stimulus and spike-triggered stimulus distribution [16,17]. However, this method does not incorporate joint information between the mean and the variance, and there is also no specific measurement for selecting the most significant subspaces based on that information. We calculated the most important eigenvectors of the STC matrix underlying the mixed stimulus and neural response (see Figure 1). As one can expect, the first eigenvector of the STC matrix resembled that obtained using the iSTAC method and was similar to the STA of the asynchronous spikes. Unlike the first eigenvector of the STC matrix, the second eigenvector comprised both slow and fast features of the stimulus. Therefore, the 2D projection of the spike-triggered stimulus matrix on the eigenvectors of the STC matrix cannot be clearly separated into two distinct clusters.

To avoid this problem, we used the iSTAC method, which allows us to choose the eigenvectors of the spike-triggered stimulus ensemble matrix more precisely by minimizing the Kullback–Leibler (KL) divergence between the eigenvectors of this matrix and that obtained using raw stimulus distributions [17]. It is to be noted that the whitening transformation is usually used before finding subspaces of the spike response. One can use the whitening transformation to calculate the uncorrelated and normalized subspaces (for both STC and iSTAC methods), which represent an unbiased estimate of the neurons’ temporal features. However, due to the type of mixed stimulus (i.e., structured and not a random process), we found that eliminating this transformation results in more representative subspace features, as shown in Figure 1, but it should be noted that these subspace features are biased to the choice of input stimuli. We compared the 2D projection of the spike-triggered stimulus matrix on the eigenvectors of the iSTAC method with and without whitening. It can be clearly observed that the iSTAC without whitening can better separate the 2D space.

### 3.2. Choice of Static Nonlinearity in the LNL Model

The static nonlinearities obtained in the augmented LNL model explains why the synchronous and asynchronous spikes are associated with different functions. For example, the smoothness and linear behavior of $f_{Async}\left(x\right)$, for *x* > 0, generates a smooth PSTH for the asynchronous spikes, which linearly encodes to the intensity of the stimulus. In contrast, the sigmoid-like nonlinearity of the synchronous spikes, $f_{Sync}\left(x\right),$ maintains the sparse PSTH of synchronous spikes because thanks to the steep nonlinearity. It is worth mentioning that more flexible nonlinear functions could provide better fits for representing the PSTH of the synchronous and asynchronous spikes. Of note, one can use deep neural networks (DNNs) to give more flexibility to the models. A DNN is simply a high-dimensional non-linear function estimator that gives a multilayer nonlinear function in the form of a neural network [25,26]. However, the main challenge with DNNs is their large parameter set, which demands a relatively larger dataset compared to what we used here to optimize the model parameters. Not only that, but also due to the large degree of freedom given by a DNN, we lose the interpretability of the modeling mechanism. In cases where these two points are concerned, one can replace our augmented LNL with a DNN to carry out the multiplexed encoding.

### 3.3. Generalized Linear Model (GLM) for Augmented LNL

The proposed augmented LNL can also be interpreted in the GLM framework. From this point of view, synchronous and asynchronous PSTHs are modeled with two separated GLMs with different random processes that eventually combine their PSTHs linearly. The first GLM filters the mixed stimulus with the first eigenvector of iSTAC and then, by passing it through a nonlinearity and then a Gaussian random process (with a linear nonlinearity as its link function), it models the PSTH related to the asynchronous spikes. Likewise, the second GLM filters the mixed stimulus with the second eigenvector of iSTAC and then, by passing it through a nonlinearity and then a Bernoulli random process (with a sigmoid nonlinearity as its link function), it models the PSTH related to the synchronous spikes (see Methods for more details about GLMs). Alternatively, we could simply interpret the augmented LNL as a single Poisson GLM with two input filters (the first two eigenvectors of iSTAC) and a Poisson random process at the end (see Appendix A for more details).

To reach the optimal parameter set for the model and avoid computational complexity, we use simple parametric models for the static nonlinearities [9]. We also can make the model more flexible by considering a flexible link function. We expect this to lead to a better fit for our model when using more complex models of neurons and eventually to lead to a better performance in encoding the fast and slow signals. For example, we can use a more flexible parametric function (with parameter set θ) such as the ex-quadratic function $f_{\theta }\left(x\right)$ as the static nonlinearities. To use the ex-quadratic function as nonlinearities we eventually need to optimize a convex cost function, which gives the optimum parameter set θ for the nonlinearity and can be optimized using a maximum-likelihood (ML) algorithm (details in Appendix B) [9]. The downside of this modeling is its computational complexity and the harder interpretation of the resulting model due to more parameters used in comparison with what we showed in this research.

## 4. Materials and Methods

### 4.1. Stimulated Mixed Input

According to the feasibility of neural systems to multiplexed coding, we simulated the activity of a homogeneous neural ensemble in response to a mixed stimulus to explore how much information can be encoded by different patterns of spikes. Each neuron received a mixed signal $\left(I_{mixed}\right)$ that consists of a fast signal $\left(I_{fast}\right)$ and a slow signal $\left(I_{slow}\right)$. $I_{fast}$ stands for the timing of the fast events or abrupt changes in the stimulus and was generated by convolving a randomly (Poisson) distributed Dirac-delta function with a synaptic waveform (normalized to the peak amplitude), $\tau_{rise}=0.5\;\mathrm{ms}$ and $\tau_{fall}=3\;\mathrm{ms}$. The fast events occurred at a rate of ∼1 Hz and were scaled by $a_{fast}=85\;\mathrm{pA}$.

$I_{slow}$ was generated using an Ornstein-Uhlenbeck process as follows.
(3)$$\frac{dI_{slow}}{dt}=-\frac{I_{slow}\left(t\right)-\mu }{\tau }+\sigma \frac{\sqrt{2}}{\tau }\xi \left(t\right),\;\;\xi ∼\;\left(0,1\right)$$
where ξ is a random number drawn from a Gaussian distribution and τ=100 ms is the time constant of the slow signal that produces a slow-varying random walk with an average of µ=15 pA and a standard deviation of σ=60 pA. The mixed signal $\left(I_{mixed}\right)$ was obtained by adding $I_{fast}$ and $I_{slow}$, which were generated independently.

An independent noise (equivalent to the background synaptic activity) was added to each neuron; thus, each neuron receives a mixed signal plus noise. Similarly, the noise ($I_{noise}$) was generated using an Ornstein-Uhlenbeck process of τ=5 ms, µ=0 pA, and σ=10 pA.

### 4.2. Simulated Neural Ensemble and Its Response to the Mixed Input

The neural ensemble consists of 30 neurons, each of which was modeled with Morris–Lecar equations [13,27]. The equations of a single model neuron receiving a mixed signal plus noise can be written as follows.
(4)$$\begin{array}{cc} C\frac{dV}{dt}=I_{mixed}\left(t\right) & +I_{noise}\left(t\right)-{\bar{g}}_{Na}m_{\infty }\left(V\right)\left(V-E_{Na}\right)-{\bar{g}}_{K}w\left(V-E_{K}\right)-g_{L}\left(V-E_{L}\right)\\ & -{\bar{g}}_{AHP}z\left(V-E_{K}\right)-g_{exc}\left(V-E_{exc}\right)-g_{inh}\left(V-E_{inh}\right) \end{array}$$
where
(5)$$\frac{dw}{dt}=\phi \frac{w\left(V\right)-w}{\tau_{W\left(V\right)}}$$
(6)$$\frac{dz}{dt}=\frac{\frac{1}{1+e^{(\beta_{z}-V)/\gamma }\;}-z}{\tau_{z}}$$
(7)$$m_{\infty }\left(V\right)=0.5\left[1+tanh\;\left(\frac{V-\beta_{m}}{\gamma_{m}}\right)\;\right]$$
(8)$$w_{\infty }\left(V\right)=0.5\left[1+tanh\;\left(\frac{V-\beta_{w}}{\gamma_{w}}\right)\;\right]$$
(9)$$\tau_{w}\left(V\right)=\frac{1}{cosh\;\left(\frac{V-\beta_{w}}{2\beta_{w}}\right)\;}$$
where $\left\{{\bar{g}}_{NA}=20,\;{\bar{g}}_{k}=20,\;{\bar{g}}_{L}=20,\;{\bar{g}}_{AHP}=25,\;g_{exc}=1.2,\;g_{inh}=1.9\right\}\frac{\mathrm{mS}}{{\mathrm{cm}}^{2}}$, $\beta_{m}=-1.2\;\mathrm{mV},\;\;\gamma_{m}=18\;\mathrm{mV},\;\;\beta_{w}=-19\;\mathrm{mV},\;\;\gamma_{w}=10\;\mathrm{mV},\;\;\beta_{z}=0\;\mathrm{mV},\;\;\gamma_{z}=2\;\mathrm{mV}$,$\tau_{a}=20\;\mathrm{ms},\;\phi =0.15,\;\mathrm{and}\;C=2\frac{\mu F}{{\mathrm{cm}}^{2}}$. These parameters were set to ensure that a neuron operates in a hybrid mode [28], i.e., an operating mode between integration and coincidence detection [5,29]. The surface area of the neuron was set to $200\;{µm}^{2}$ so that $I_{mixed}$ is reported in pA, rather than as a density [30,31]. Figure 1A shows the mixed stimulus and the spiking activities of the ensemble of neurons in response to this stimulus.

### 4.3. Generalized Linear Model (GLM) Details

A GLM is a generalization of traditional linear models that gives the neural encoding models more flexibility to capture the nonlinear dynamics of neural activity. GLMs contain three stages. The first stage is a linear mapping that consists of a set of d linear filters. Let us assume $K=\left[k_{1},\ldots ,k_{D}\right]$, which maps a high-dimensional sensory stimulus $s\in R^{M}$ onto a low-dimensional stimulus feature map $x\;\in R^{D}$:(10)$$x=K^{T}s$$

The second stage is a pointwise nonlinearity, $f:\;R^{D}\to R$, which maps the linear features of d dimensions into a nonnegative spike rate:(11)$$\lambda =f\left(x\right)$$In the final stage, the number of spikes is generated by a random process:(12)$$P_{\theta }(Y=r|s)\;$$
where Y is a random variable related to spike occurrence, r is the instantaneous firing rate, and θ is the parameter set of the random process

In simple words, by using a GLM we approximate the instantaneous firing rate by considering features from D dimensions instead of M dimensions:(13)$$P(Y|s)∼P(Y|K^{T}s)$$

Thus, there are two sets of parameters, the estimators (K) and the pointwise nonlinearity (f), which can be optimized to reach the desired model.

### 4.4. STA and STC Estimators

If we assume that $p\left(s\right)$ has zero mean, then the STA can be defined as the average of the stimulus given the instantaneous firing rate:(14)$$\mu =\frac{1}{n_{sp}}\sum_{\left\{s_{i}|spike\right\}}^{}s_{i},\;n_{sp}=\sum_{t=1}^{N}r_{t}$$
where N is the total number of time points. The STA is an unbiased, consistent estimation that gives the direction in the stimulus space along which the means of P(s|spike) and $P\left(s\right)$ differ the most. The problem is the STA gives a single direction in stimulus space and leads to a single estimator, which is not sufficient to capture all the information in the stimulus space (as previously discussed, we have a mixed stimulus in this research). To consider other possible directions with maximal differences in the variances between P(s|spike) and $P\left(s\right)$ we can use the eigenvectors of the STC matrix, defined as:(15)$$\Lambda =\frac{1}{n_{sp}}\sum_{\left\{s_{i}|spike\right\}}^{}\left(s_{i}-\mu \right){\left(s_{i}-\mu \right)}^{T}$$

The STA and eigenvectors of the STC matrix can provide a basis, K, for a reduced-dimensional model of the neural response.

### 4.5. iSTAC Estimator

There are major problems with STA/STC. The measure we use to select the eigenvectors of STC is based on eigenvalues, which do not truly represent the most informative directions. As we mentioned before, the objective in iSTAC is to reduce the KL divergence between the Gaussian approximations to the spike-triggered and raw stimulus distributions. Therefore, we define Q as a Gaussian approximation of P(s|spike) based on the information contained only in the mean and covariance of P as:(16)$$Q\left(s\right)=\frac{1}{{\left(2\pi \right)}^{\frac{n}{2}}{\left|\Lambda \right|}^{\frac{1}{2}}}\;e^{-\frac{1}{2}{\left(s-\mu \right)}^{T}\Lambda^{-1}{\left(s-\mu \right)}^{}}$$
where *n* is the dimensionality of the stimulus space. Now we drive KL the divergence between Q and P as:(17)$$D\left(Q,P\right)=\int Q\left(s\right)\;\log\left(\frac{Q\left(s\right)}{P\left(s\right)}\right)\;ds$$

By considering that the mean of P and P∘Q is zero and have identity covariance (if not, we can use a “whitening” technique) we can rewrite D in a simpler form as:(18)$$D\left(Q,P\right)=\frac{1}{2}\left(Tr\left(\Lambda \right)-\log\left|\Lambda \right|+\mu^{T}\mu -n\right)$$
where $Tr\left(.\right)$ and $\left|.\right|$ indicate the trace and determinant, respectively.

Based on the fact that we are interested in d subspaces we can approximate the D with:(19)$$D_{\left[K\right]}\left(P,Q\right)=\frac{1}{2}\left(Tr[K^{T}\left(\Lambda +\mu^{T}\mu \right)K-loglog\;\left|K^{T}\Lambda \;K\right|-d]\;\right)$$
where d is the dimension of the interested subspaces.

Thus, in terms of finding the d most informative subspaces decomposed by STA and the eigenvectors of STC, we need to find $D_{\left[K\right]}\left(P,Q\right)$ for all subspaces and select the first d ones.

An important advantage of the iSTAC approach over traditional STA/STC analysis is that it makes statistically efficient use of changes in both the mean and covariance of the response spaces.

## Figures and Tables

**Figure 1 entropy-25-00589-f001:**
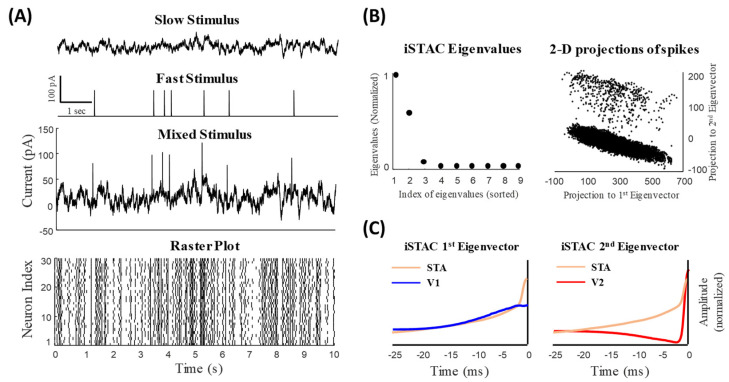
Slow and fast features of a mixed signal can be inferred from responses of a homogeneous ensemble of neurons using the iSTAC method. (**A**) Slow and fast signals comprising a mixed signal. (Bottom) Sample raster plot of 30 model neurons receiving the common mixed signal (and independent noise). Spikes evoked by the fast and slow signals cannot be distinguished visually. (**B**) The iSTAC method was applied to spike-triggered mixed signal and eigenvalues and eigenvectors were obtained (see Methods). (**Left**) The eigenvalues of the iSTAC matrix reveal two significant components of the population code. (**Right**) The projection of spike-triggered mixed signal onto the main eigenvectors of the iSTAC matrix. Two clusters can be visually distinguished. (**C**) The 1st and 2nd eigenvectors of the iSTAC matrix, *V*1 and *V*2, respectively, are shown against the spike-triggered average (STA). *V*1 resembles the STA filter reflecting slowly-varying changes in the signal. Unlike *V*1, *V*2 resembles a high-pass filter (differentiator) that reflects fast features of the mixed signal.

**Figure 2 entropy-25-00589-f002:**
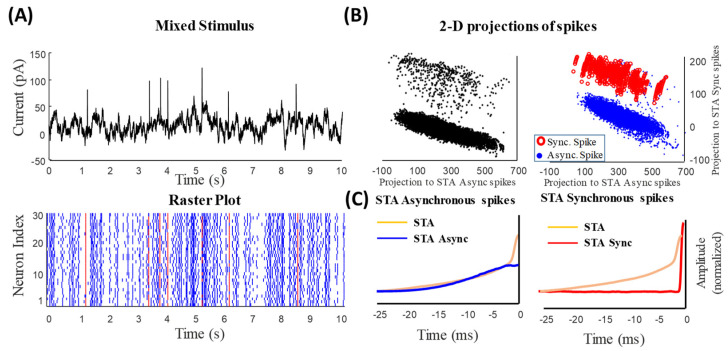
Synchronous and asynchronous spikes represent slow and fast features of the mixed signal, respectively. (**A**) Synchronous (red) and asynchronous (blue) spikes are distinguished by setting a threshold on the instantaneous firing rate calculated by a narrow kernel (see Methods). Synchronous spikes evoked by the fast signals can be distinguished visually. (**B**) The projection of spike-triggered mixed signal onto the STA_Sync_ and STA_Async_. Two (visually) distinguishable clusters belong to asynchronous spikes representing the slow feature of the signal (blue dots) and synchronous spikes representing the fast features (red circles). (**C**) The spike-triggered average of synchronous (red) and asynchronous (blue) spikes, namely, STA_Sync_ and STA_Async_, respectively, is shown against the STA of all spikes (similar to Figure 1C).

**Figure 3 entropy-25-00589-f003:**
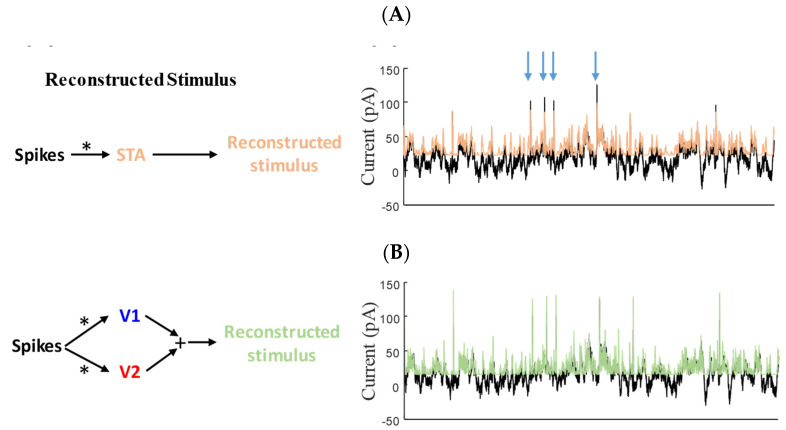
Block diagram of decoding the mixed signal from spikes using (**A**) the STA filter (light brown), (**B**) a weighted sum of the 1st and 2nd eigenvectors of the iSTAC method (green), and (**C**) a weighted sum of filtered asynchronous spikes (by STA_Async_) and filtered synchronous spikes (by STA_Sync_) (purple). Original mixed signal (black) is overlaid with reconstructed signal (color) in the plots. As can be seen in these figures, the reconstructed signal based on STA_Sync_ and STA_Async_—similar to that obtained by eigenvectors of iSTAC method—can capture both slow and fast components of the signal accurately. “*” indicates the spiking signals.

**Figure 4 entropy-25-00589-f004:**
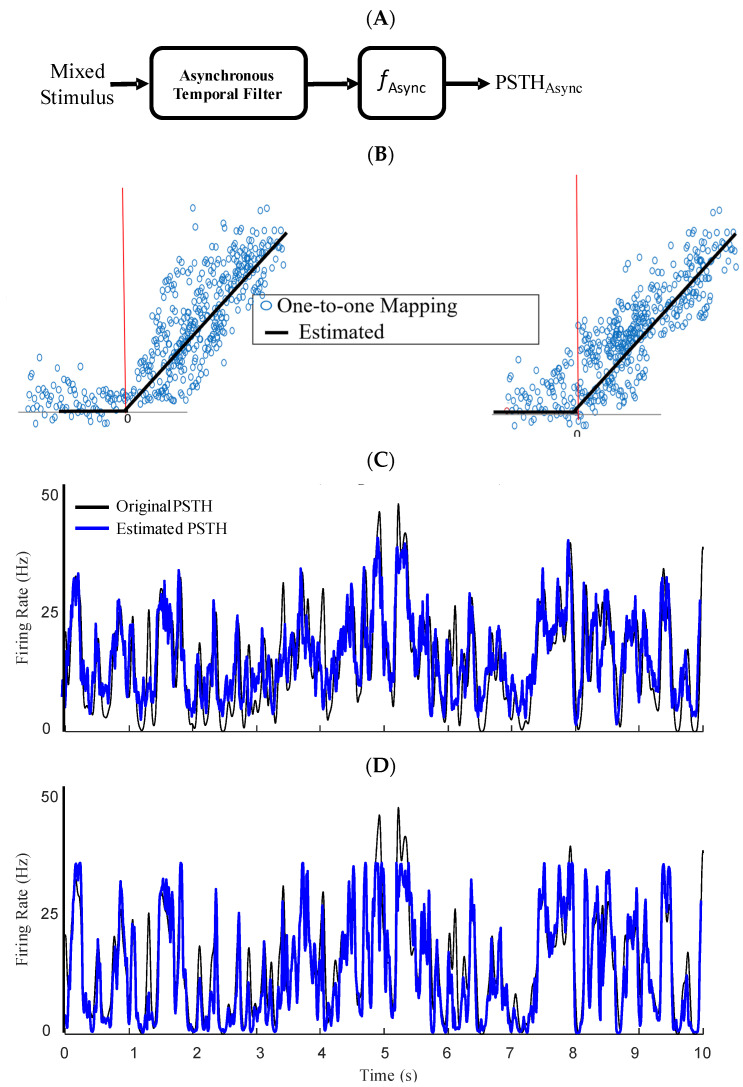
Static nonlinearities underlying asynchronous spikes. (**A**) Block diagram of LNL model for asynchronous spikes. (**B**) Static nonlinearity calculated for asynchronous spikes is obtained by mapping the output of filtered stimulus to the instantaneous rate of asynchronous spikes (calculated with a wide kernel, σ = 25 msec). Static nonlinearity calculated based on 1st eigenvectors of the iSTAC method, $v_{1}$, (**Left**) and STA_Async_ (**Right**). The solid black shows fitted rectifiers. (**C**) The PSTHs constructed using the fitted nonlinearities based on $v_{1}$ were drawn against the PSTH of asynchronous spikes. (**D**) The PSTHs constructed using the fitted nonlinearities based on STA_Async_ were drawn against the PSTH of asynchronous spikes.

**Figure 5 entropy-25-00589-f005:**
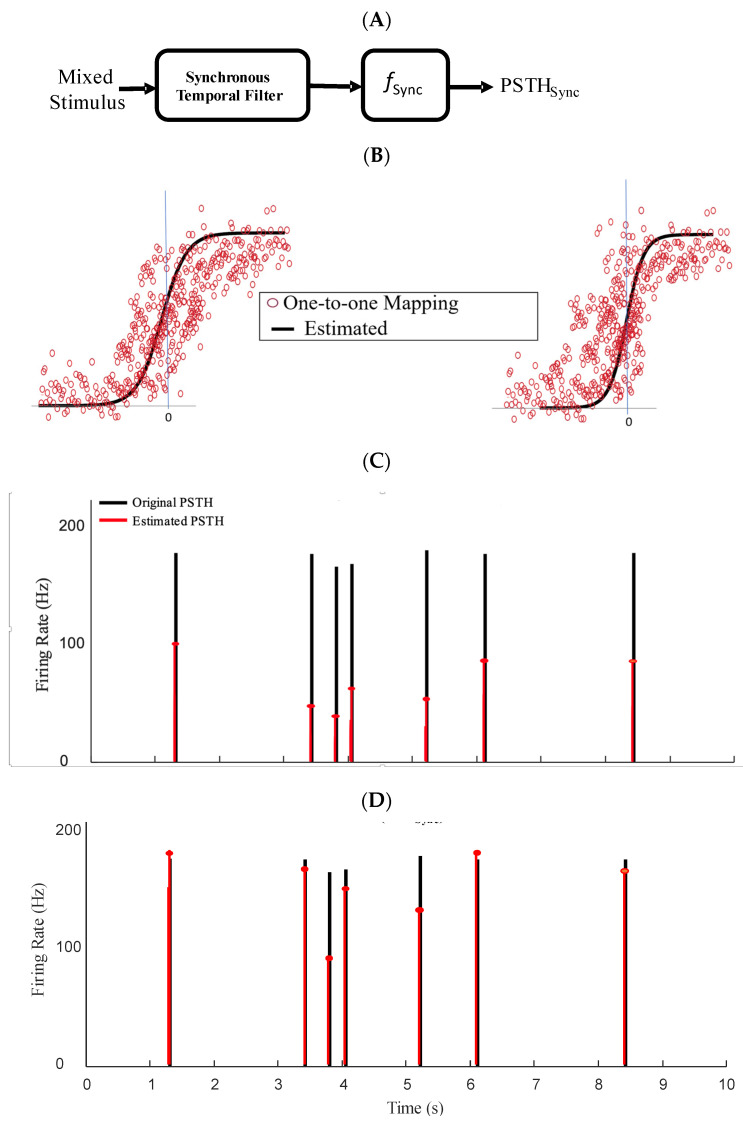
Static nonlinearities underlying synchronous spikes. (**A**) Block diagram of LNL model for synchronous spikes. (**B**) Static nonlinearity calculated for the synchronous spikes is obtained by mapping the output of filtered stimulus to the instantaneous synchronous events (calculated with a narrow kernel, σ = 1 msec). Static nonlinearity calculated based on 2nd eigenvectors of the iSTAC method, $v_{2}$, (**Left**) and STA_Sync_ (**Right**). The solid black shows fitted sigmoid functions. (**C**) The PSTHs constructed using the fitted nonlinearities based on $v_{2}$ were drawn against the PSTH of synchronous spikes. (**D**) The PSTHs constructed using the fitted nonlinearities based on STA_Sync_ were drawn against the PSTH of synchronous spikes.

**Figure 6 entropy-25-00589-f006:**
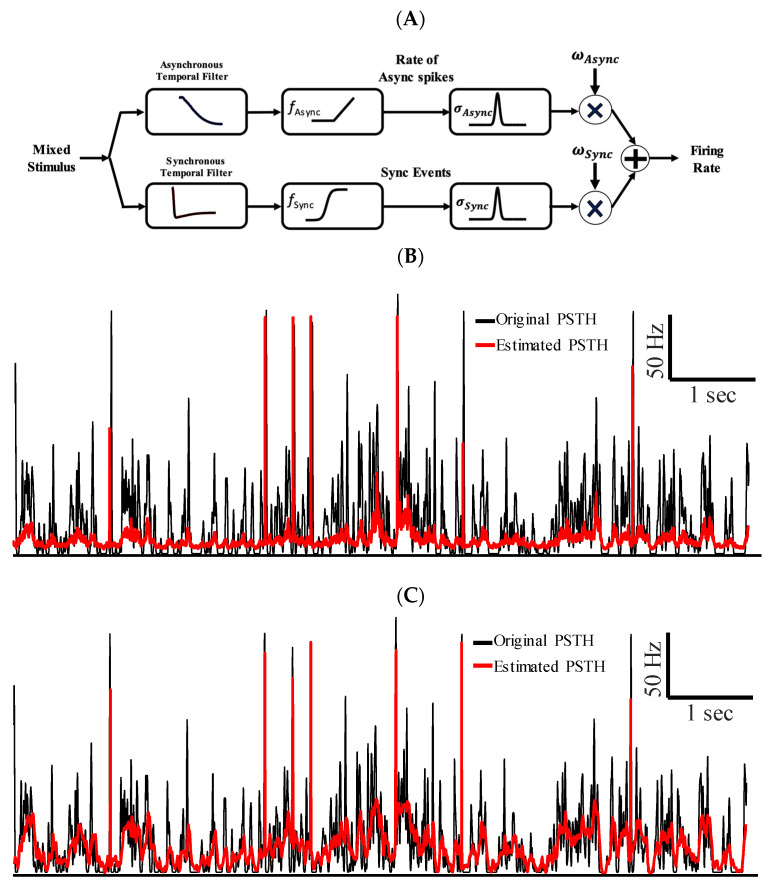
Two-stream LNL model, referred to as augmented LNL model, enables co-existence of temporal and rate codes. (**A**) Block diagram of the augmented LNL model for combining rate of asynchronous and synchronous spike events. (**B**) The PSTHs estimated using a conventional Poisson GLM (red) are shown against the original PSTH (calculated with a 1 msec Gaussian kernel). (**C**) The PSTHs estimated using the segmented LNL using temporal filters of iSTAC method. (**D**) The PSTHs estimated using LNL using the segmented LNL using STA_Async_ and STA_Sync_.

**Table 1 entropy-25-00589-t001:** Mean absolute error (MAE) and root mean squared error (RMSE) performance measure comparison between Poisson GLM and augmented LNL model on test data.

	(MAE)			RMSE		
	Sync	Async	Mixed	Sync	Async	Mixed
Poisson GLM	0.245	0.223	0.228	0.338	0.313	0.326
Augmented LNL ($STA_{V_{1},}\;STA_{V_{2}}$)	0.101	0.104	0.102	0.137	0.134	0.135
Augmented LNL ($STA_{ASync}\;,STA_{Sync}$)	0.103	0.107	0.106	0.140	0.151	0.149

## Data Availability

The data and suggested model here is available at https://github.com/MrRezaeiUofT/Synchrony-Division-Neural-Multiplexing.git.

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
