# Peer review of "Synchrony-Division Neural Multiplexing: An Encoding Model"

_entropy, 2023, doi:10.3390/e25040589_

Round 1

Author Response

Reviewer 1: Line 88: How will the number of neurons used in the simulation affect the results

Response: It was based on our previous work [PNAS 2019 - PMID 31028148]. We showed that 30 neurons are necessary to detect synchrony and to consider the instantaneous firing rate as a reliable code.

Line 91: What is the hybrid mode and how does it affect the resulting filters?

Response: The hybrid model is referred to the properties of single neurons to perform filtering. For example, for specific parameters of conductances in the model neuron it can behave as a low pass filter (integrators like slow adapting sensory neurons) or a high pass filter (coincident detectors like fast adapting neurons). A hybrid mode refers to a band-pass filter where the neuron is not a pure integrator or a pure coincident detector.

Figure 4B and 5B: What are the colored lines? This visualization looks confusing. 

Response: We appreciate the reviewer’s point. Please note that the colored lines in Figures 4B and 5B represent the mapping from the filtered signal to the firing rate, for all samples. We re-plotted these figures and considered “markers” to enhance their visualizations.

Line 231-234: Attributing the better performance of a higher dimensional model versus that of a simpler GLM (with less number of dimensions and parameters) to the significance of the components of the more complex model is not a valid conclusion.

Response: We fully agree with the reviewer’s point. Although the two-stream linear nonlinear (LNL) model outperformed a simple GLM, this comparison is not fair as these models have different free parameters. We acknowledged this point in the revised version. Moreover, we notice that our main purpose here was not to compare the performance of the two-stream filter with a simpler model but to highlight the feasibility of the two-stream model (whose performance is higher than a baseline model). This comparison suggests that in addition to the mechanistic values of the two-stream LNL model, it can be used for mapping input signals to spikes (similar to a simple GLM model).

Figure 3-6: Demonstrating or comparing the performance of different models only via visual inspection and using a sample chunk of input or output data is not sufficient for a modeling paper. Quantitative analysis at a population level needs to be provided.

Response: Thank you for mentioning this. We compared the performance of Poisson GLM and our augmented LNL with mean absolute error (MAE)  and root mean squared error (RMSE) measures for the test data in Table 1.

It is not clear if the decoding performance in Fig. 3 or the prediction performance in Fig. 4-6 have been measured using test data (not used for the model fitting) or using training data. The former is expected to be used for these analyses

Response: Thank you for bringing this up. Yes, the result in figs 4-6 are measured using test data. We selected the first 50% of our simulated data set for training and the remaining 50% for the test. We clarified that on the paper.

Line 290-294: For the same reason mentioned here, unless a whitening process is applied, the obtained filters (models) cannot represent an unbiased estimate of the neurons’ temporal features. The filters are biased for the choice of input stimuli. Clarify this in the results section. 

Response: Thank you for mentioning this, we clarified it in the paper.

Minor edits: Line 144-146: ‘respectively’ does not seem to reflect the correct order of relationships

Response: Thank you for mentioning it we fixed the ordering.

Figure 4A and 5A caption: replace ‘blog diagram’ with ‘block diagram’.

Response: We fixed the typo.

Figure B title: this is supposed to be STA sync.

Response: Thank you, we fixed this.

Line 280: section 2b does not discuss the results of the STC method. 

Response: Yes the result of this discussion is provided in Figure 1. We fixed it in the text.

Line 307-311: The DNN discussion here does not sound relevant to the purpose of this study. 

Response: We rephrased the section, identified by red color in the revised manuscript, to clarify our massage here.

Section 4.c.2: Why is ‘log log’ used instead of ‘log’?

Response: It should be only one ‘log’ we fixed that in the manuscript.

Line 398: This is not an accurate statement. STC can also be applied to the stimulus space orthogonalized to the STA.

Response: Thank you for mentioning this. We fixed this. 

Overall the presented results can be useful in terms of providing another example of how simple linear nonlinear cascade models are able to capture various neural codes. Thus the work can be of interest to a relatively broad audience in the field of functional modeling.

Response: Thank you. Yes, it could be useful in a broader field of functional modeling.

Reviewer 2 Report

The authors develop novel inference tools to decode mixed stimulus from spiking data and further train temporal filters to predict firing rates from stimulus data. The computational framework has been rigorously evaluated, and the paper is well-written. I recommend publication with minor revisions. 

  1. I did not find the description of the kernel to distinguish synchronous and asynchronous spikes in Methods. Is it at lines 222-223?
  2. PSTH is not defined. 
  3. Typos "loglog" in Eq. 17-19.
  4. Can the authors explain why the reconstructed stimulus matches poorly with data close to or below zero in Fig 3? 

Author Response

Reviewer 2: I did not find the description of the kernel to distinguish synchronous and asynchronous spikes in Methods. Is it at lines 222-223?

Response: line 222-223 discusses the smoothing kernel selected to promote smoothness in the output. The kernels are shown in Figure 4.B and 5.B for synchronous and asynchronous spikes, respectively. We clarified it in the manuscript. 

PSTH is not defined.

Response: Thanks for mentioning it, we define the acronym. 

Typos "loglog" in Eq. 17-19.

Response: we fixed it. 

Can the authors explain why the reconstructed stimulus matches poorly with data close to or below zero in Fig 3?

Response: We believe this happened due to the selection of link functions, as you can see we have sigmoid and Relu nonlinearities which map values less than zeros to zero, and az a result it is not able to accurately reconstruct data below zero. 

Reviewer 3 Report

Overview of the manuscript
The work focuses on presenting the computational framework procedure to provide a system to analyse how an ensemble of homogeneous neurons enable such synchrony division multiplexing (SDM). The authors used a conductance-based model of an ensemble of homogeneous  neurons receiving a mixed stimulus comprising slow and fast features.

Using feature estimation techniques, the authors evidenced that both features of the stimulus can be inferred from the generated spikes.  Furthermore, the authors using linear nonlinear (LNL) cascade models and calculate temporal filters and static nonlinearities of differentially synchronized spikes, demonstrating that filters and nonlinearities computational model were distinct for synchronous and asynchronous spikes.  Finally, the authors developed an augmented LNL cascade model as an encoding model for the SDM by combining individual LNLs calculated for each type of spike, noting that a homogeneous neural ensemble model can simultaneously perform temporal- and rate- coding function.

GENERAL COMMENT

The work is interesting, it highlights an interesting computational procedure to analysing the spiking response of neurons. The methodology is rigorous, widely described and well-focused on the computational procedure to evidence the multiplexing properties of an ensemble of neurons. The results are adequately presented and well support the conclusions. However, the Discussion section deserves more attention. Also, the paragraph “Subspace feature extractors; iSTAC vs STC” detailed in Discussion section does not have an adequate presentation in Results section.

Detail the acronyms as they first appear.

SPECIFIC COMMENTS

Abstract

Pag. 1, line 29: “a homogeneous neural ensemble” model. May be better.

 Results

Legend Fig. 3B: “light purple”? It is green, isn’t it?

Pag. 7, line 186: explicit the acronym.

Pag. 8, line 195: explicit the acronym.

Discussion

Pag. 11-12, line 244-245: the sentence remains not clear. Rephrase it.

Pag. 12, line 267-285: In this paragraph you discuss about iSTAC vs STC, but in Results section this topic is not presented. Similarly, you indicate “see section 2.b”, but the section deals with space estimator STA. Explain this gap or resolve it.

Pag. 13, line 309-311: The conclusion is not clear. You mean that the use of DNN is the same of your use of augmented LNL model. If it so, why you have not used. Explain better your conclusion.

Pag. 14, line 327-331: What is the point? It is not clear your conclusion. Explain better.

 Materials and Methods

Pag. 14, line 342: explain the acronym

Pag. 15, line 361-362. The sentence is useless, here you describe methods. Delete the sentence.

Author Response

Reviewer 3: Pag. 1, line 29: “a homogeneous neural ensemble” model. May be better.

Response: Yes, we fixed that.

Legend Fig. 3B: “light purple”? It is green, isn’t it?

Response: Yes, we fixed that.

Pag. 7, line 186: explicit the acronym.

Response: we fixed that.

Pag. 8, line 195: explicit the acronym.

Response: we defined it in line 186

Pag. 11-12, line 244-245: the sentence remains not clear. Rephrase it.

Response: Thank you we fixed this.

Pag. 12, lines 267-285: In this paragraph, you discuss about iSTAC vs STC, but in Results section this topic is not presented. Similarly, you indicate “see section 2.b”, but the section deals with space estimator STA. Explain this gap or resolve it.

Response: There is a detailed discussion and analysis of iSTAC vs STC in reference [17] and also additional discussions in [23,24]. Therefore we ignore including the comparison result for these two models. Therefore, this paragraph tries to motivate the importance of iSTAC and its advantage to the STC. We fixed “see section 2.b” and it should be Figure 1.

Pag. 13, line 309-311: The conclusion is not clear. You mean that the use of DNN is the same of your use of augmented LNL model. If it so, why you have not used. Explain better your conclusion.

Response: We rephrased the section, identified by red color in the revised manuscript, to clarify our massage here.

Pag. 14, line 327-331: What is the point? It is not clear your conclusion. Explain better.

Response: The point here is to keep our model simple and interpretable we used simple parametric functions for the nonlinearity part of our model. We also can make the model more flexible by considering a flexible link function. We expect this leads to a better fit for our model in case of using more complex models of neurons and eventually leads to a better performance in encoding the fast and slow signals. The downside of this modeling is computational complexity and harder interpretation of the resulting model due to more parameters we used in comparison with what we showed in this research.

Pag. 14, line 342: explain the acronym

Response: it is an Euler process, we fixed that

Pag. 15, line 361-362. The sentence is useless, here you describe methods. Delete the sentence.

Response: we deleted it.

Round 2

Reviewer 1 Report

The authors' responses to my comments and the revision sufficiently addressed my concerns.

A minor typo to correct: duplicate word 'mapping' in the legend of Fig.4B and Fig. 5B

Reviewer 2 Report

Thanks for the authors' reply. I recommend the paper for pubilication.

Reviewer 3 Report

No more concerns.